# Optimal Design of Ferronickel Slag Alkali-Activated Material for High Thermal Load Applications Developed by Design of Experiment

**DOI:** 10.3390/ma15134379

**Published:** 2022-06-21

**Authors:** Andres Arce, Anastasija Komkova, Jorn Van De Sande, Catherine G. Papanicolaou, Thanasis C. Triantafillou

**Affiliations:** 1Department of Civil Engineering, University of Patras, GR-26504 Patras, Greece; kpapanic@upatras.gr (C.G.P.); ttriant@upatras.gr (T.C.T.); 2Department of Civil, Environmental and Geomatic Engineering, ETH Zurich, 8093 Zurich, Switzerland; komkova@ibi.baug.ethz.ch; 3Department of Materials and Chemistry, Vrije Universiteit Brussel, 1050 Brussels, Belgium; jorn.van.de.sande@vub.be; 4Department of Materials Engineering, KU Leuven, 3001 Leuven, Belgium

**Keywords:** alkali-activated materials (AAM), design of experiment (DOE), life cycle assessment (LCA), response surface method (RSM)

## Abstract

The development of an optimal low-calcium alkali-activated binder for high-temperature stability based on ferronickel slag, silica fume, potassium hydroxide, and potassium silicate was investigated based on Mixture Design of Experiment (Mixture DOE). Mass loss, shrinkage/expansion, and compressive and flexural strengths before and after exposure to a high thermal load (900 °C for two hours) were selected as performance markers. Chemical activator minimization was considered in the selection of the optimal mix to reduce CO_2_ emissions. Unheated 42-day compressive strength was found to be as high as 99.6 MPa whereas the 42-day residual compressive strength after exposure to the high temperature reached 35 MPa (results pertaining to different mixes). Similarly, the maximum unheated 42-day flexural strength achieved was 8.8 MPa, and the maximum residual flexural strength after extreme temperature exposure was 2.5 MPa. The binder showed comparable properties to other alkali-activated ones already studied and a superior thermal performance when compared to Ordinary Portland Cement. A quantitative X-ray diffraction analysis was performed on selected hardened mixes, and fayalite was found to be an important component in the optimal formulation. A life-cycle analysis was performed to study the CO_2_ savings, which corresponded to 55% for economic allocation.

## 1. Introduction

The world is currently seeking to reduce and, if possible, minimize carbon emissions. Each year, the construction sector consumes approximately 1.6 billion tons of cement, which translates to the release of 1.28 billion tons of CO_2_ [1]. The cement industry is the second-largest producer of carbon emissions, amounting to 5–8% of total global emissions [2]. While Portland cement is—and will be for many years—the primary material of choice for construction, there is a growing concern regarding the sustainability of this material [3], and thus, there is a need for investigating alternative, more sustainable binders. A low-calcium alkali-activated material (AAM) comprises an aluminosilicate source that is low in calcium and an alkaline solution. This type of material has the potential to develop high mechanical strength and excellent fire and chemical resistance [4]. Low-calcium AAM technology also has the potential to reduce carbon emissions by 80% [5] compared to OPC. The increase in the popularity of low-calcium alkali-activated materials in the last decades can be explained by their ability to consume readily available industrial by-products, with the consequence of reducing problems related to by-products storage and the shortage of natural resources [6].

The exposure of an OPC-based paste to high thermal loads (higher than 330 °C) results in the degradation of its mechanical properties due to changes in the chemical composition of the hydration products and micro-cracks produced by differential thermal stresses in the matrix. The deterioration of chemical compounds and bonds occurs as calcium hydroxide groups begin to decompose between 330 °C and 400 °C. Calcium carbonate decomposes at 700 °C and melts at 800 °C [7]. The main difference and advantage in the fire resistance of low-calcium AAM is the absence or low presence of C-S-H gel. The decomposition of this gel after 300 °C is the primary reason for mechanical strength degradation in OPC-based products [8,9,10]. Zhang et al. [3] found that the compressive strength of hardened OPC paste after exposure to 800 °C for 1 hour was completely lost. An identical outcome was reported by Mendes et al. [11], who subjected OPC paste cylinders to 750 °C for 1 h followed by 800 °C for an additional hour. For low-calcium AAM, the residual strength after a fire depends on the precursor and activator solutions, with the results varying greatly. After exposure to the same temperature, Kong et al. [12] reported a residual compressive strength of 49% of the initial one for a metakaolin-based low-calcium AAM. Kong and Sanjayan’s [13] results showed an increase in compressive strength of 53% after similar high-temperature treatment. Martin et al. [14] reported higher mechanical strength and better fracture performance for alkali-activated materials when compared to OPC-based ones after a high thermal load (HTL). 

Ferronickel slag (FNS) is a by-product of the production process of ferronickel alloy. The process is carried out in an electric arc furnace, which reaches a temperature of 1300–1500 °C. The liquid slag flows out of the furnace at high temperatures and is led over a seawater jet under pressure. There, it is instantly turned into finer particles which are cooled and transported by the water stream to a collection pond. Because of the instantaneous cooling to ambient temperature, the slag comes practically in an amorphous phase. The granules’ size is under 4 mm, with a prevailing size between 0.6 and 1.5 mm [15]. The yearly production of ferronickel slag was approximately 2 million tons for Greece (reported in 2022 [16]), and only 20–30% of this amount is being put back into the economy by using it as a sandblasting material and raw ingredient for cement production [17]. The remaining slag must be disposed of in surface locations or under the sea. The cost of disposal was reported to be EUR 650,000 per year in 2007 [18]. A sustainable production model for the metal industry necessitates action in the management of the residual slags. It is essential for this industry to find applications for ferronickel slag [17]. 

Ferronickel slag is only an example of a type of metallurgical slag produced every year from the ferrous, nonferrous, and steel industries. Millions of tons of these slags are disposed of on/in land, causing several environmental impacts [18,19]. Thus, the creation of a circular economy-based scheme that valorizes them and develops new applications of added value is necessary to improve the sustainability of the metallurgical sector and minimize environmental damage. This study, like others before [20,21,22], represents an effort to put these million tons of slag back into the economy. The focus is given to analyzing the potential of FNS as a primary ingredient to produce high thermal load-resistant “cement”, thus reducing the consumption of raw materials and waste allocation efforts. 

Sakkas [23] demonstrated the applicability of ferronickel slag for the development of construction materials with fire-resistance. The author reported formulations with compressive strengths as high as 120 MPa and thermal conductivities at 300 K as low as 0.27 W/ mK (not for the same formulations). This suggested that FNS AAMs have the potential for use in the development of high thermal load-resistant construction materials. The FNS mixes developed in this study belong to the group of low-calcium alkali-activated materials which are known to have high resistance to fire. Possible applications of fire-resistant low-calcium AAMs could be in highly important constructions such as tunnels, underground edifices [24], and tall structures [25]. Another advantage of low-calcium AAMs is their high resistance to chemical attacks. Komnitsas et al. [26] studied the durability of FNS AAMs and found the product to possess high resistance to corrosive environments such as sulfuric acid and chloride acid solutions after 30 days of exposure. A study of the same author on the toxicity of FNS AAM and raw FNS revealed that the raw materials’ toxicity levels exceed the allowable limit only for Ni and Cr. After alkali activation, the low-calcium AAM exhibited zero toxicity. So, FNS binders not only have potential as fire and chemical-resistant products but can be used to trap potentially hazardous elements to reduce their toxicity [27].

AAM properties are strongly dependent upon the type and content of the chemical activators and precursors. While AAMs can provide benefits such as those previously mentioned, the design of an AAM requires a delicate balance which can be achieved by employing statistical tools such as Design of Experiment (DOE). DOE is a systematical approach for determining cause-and-effect relationships. DOE is commonly used in industry for the optimization of industrial processes (Factorial DOE) or the formulation of mixes (Mixture DOE). The procedure applies to any process with measurable inputs and outputs. Until 1980, DOE was mainly used in the chemical, food, and pharmaceutical industries; in recent decades, the methodology has found a place in the concrete industry as well. This study makes use of the advantages of DOE for the formulation of low-calcium AAM binders focused on high thermal load applications and the reduction of CO_2_ emissions. This is not the first time that DOE has been used for low-calcium AAM design. In 2016, Mohd Basri et al. [28] used a factorial DOE to optimize the mechanical properties of a low-calcium AAM matrix by considering different quantities of the alkaline activator (AA), AA/precursor ratio, curing temperature, curing time, and sodium hydroxide (NaOH) concentration. While a factorial design is preferred for optimization problems with categorical variables, it is mixture DOE (the optimization of ingredient quantities varying within a range) that better qualifies for mix design optimization, and this was thus the selected approach for this study. In this type of design, a fixed amount of ingredients was selected, and only their proportions changed. Komnitsas used a DOE factorial design to optimize the compressive strength of FNS-based low-calcium AAM. Once again, this type of experiment design was focused on factors (with discrete values) such as aging, curing temperature, and curing time rather than ingredient amounts [18]. At the time this paper was written, the authors found no literature corresponding to the mixture DOE of FNS AAMs.

Mixture DOE permits the user to find the proportions of ingredients for a multi-response optimization. The ingredients used were ferronickel slag, potassium hydroxide (KOH), potassium silicate (KS), silica fume (SF), and water. The studied responses were flow spread, mass loss, shrinkage/expansion, and compressive and flexural strength before and after a high thermal load; additionally, cost and CO_2_ emissions estimates of the binder formula were considered in the selection of the optimal formula. The result was a numerically optimized formulation to produce an alkali-activated binder with a high performance, a minimal cost, and reduced CO_2_ emissions with respect to OPC alternatives. The reduction of chemical activators was key to reducing the cost and CO_2_ emissions and to making the product safer and more comfortable to work with for operators in potential construction field future applications. Considering the need for easy in situ application, it was decided to cure pastes only at ambient temperatures. 

## 2. Materials and Methods

### 2.1. Raw Materials

Ferronickel slag was selected as the main bulk component of the paste and was kindly provided by The General Mining and Metallurgical Company SA in Larissa, Greece, better known as LARCO. Ferronickel slag and silica fume were analyzed in terms of particle size distribution using laser diffraction; more specifically, the analysis was carried out using a Malvern Mastersizer 2000. The slag was ground in a ball mill to a d_50_ of 8.36 μm and a d_90_ of 29.1 μm. For the silica fume, the d_50_ and d_90_ values were equal to 12.87 μm and 29.98 μm, respectively. Potassium silicate was chosen in the form of a commercial aqueous solution called Geosil^®^ 14517 (modulus 1.6, 45% dry content). Potassium hydroxide pellets with a 90% purity were selected. Potassium products were preferred over more traditional sodium alkaline activators, as they have been reported to produce more thermally resistant low-calcium AAMs with higher compressive strengths [29]. In support of this, Kovalchuk and Krivenko [30] provided a very clear example by comparing the fusion temperature (*T_f_*) of two almost identical mineral phases: orthoclase (K_2_O·Al_2_O_3_·6SiO_2_) with *T_f_* = 1170 °C and albite (Na_2_O·Al_2_O_3_·6SiO_2_) with *T_f_* = 1118 °C, where the only difference was the alkaline metal. 

The chemical composition data of ferronickel slag and silica fume (Table 1) were gathered through X-Ray fluorescence (XRF), revealing major (SiO_2_, Al_2_O_3_, CaO, MgO, MnO, Fe_2_O_3_, K_2_O, Na_2_O, P_2_O_5_, TiO_2_) and minor elements. An amount of 1.8 g of dried ground sample was mixed with 0.2 g of wax (acting as a binder) and was pressed on a base of boric acid to a circular powder pellet that was 32 mm in diameter. Analyses were performed with a RIGAKU ZSX PRIMUS II spectrometer, which was equipped with Rh-anode running at 4 kW, for major and trace element analyses. The spectrometer was equipped with the following diffracting crystals: LIF (200), LIF (220), PET, Ge, RX-25, RX-61, RX-40, and RX-75.

Table 2 includes additional important information for each binder ingredient (referred to as “component”) such as the estimated cost (by weight of the ready-to-mix component) and the environmental impact quantified through both CO_2_ emissions (in CO_2-eq._ terms) and energy consumption for component production. The cost of the chemical activators and silica fume corresponds to the purchase of the laboratory scale purchases. The authors predict an important drop in price for a large-scale operation. The cost of the grounded was estimated to be similar to the cost of ground granulated blast furnace slag, as reported by Hendrik G [31].

The CO_2-eq._ emissions were estimated using the SimaPro v8.5 software linked to the Ecoinvent v3.4 database. In Table 2, each component was assigned a single-letter code (A to D), which will appear in figures throughout the text. Component C (KS) data always refer to the dry part (45%) of the KS solution available in the form of Geosil^®^ 14517.

### 2.2. Sample Preparation

A chemical activator solution was prepared one day in advance of paste preparation. Faucet water was added to a glass jar, followed by potassium silicate in the form of Geosil^®^ 14517; finally, potassium hydroxide pellets were dissolved in the solution and mixed with an agitator. Silica fume and ferronickel slag were dry-mixed by hand until homogenization. The activator solution was added and, once again, the paste was formed by hand-mixing the ingredients for approximately half a minute. Once the dry mix absorbed all the activator and the risk of spilling the solution was low, the mixing continued by means of an electric hand mixer for 1 min at a low speed, immediately followed by 2 min at a high speed. The consistency of the fresh paste samples was assessed by a flow table test and expressed in terms of the mean diameter of the paste after jolting the table 15 times (as per EN 1015-3). Mortar prisms of 40 × 40 × 160 mm were cast and tested after 42 days for flexural and compressive strength according to EN 1015-11. This age was deemed adequate for the mechanical characterization of the prisms aimed at a comparative study of different formulations. Once filled, the molds were vibrated for 30 s using an electric shaking table. The molds were covered with a thin plastic sheet until the paste was hard enough to demold, which was between 5 h and 24 h after mixing. The prisms were cured in two airtight plastic bags to prevent the loss of moisture. The temperature during curing was approximately 20 ± 5 °C; no heat curing was used. The specimens were tested after 42 days of curing in airtight plastic bags. 

### 2.3. Heat Treatment

All prisms programmed for heating were exposed to a thermal load comprising a ramp of 5 °C/min up to 900 °C, followed by a constant temperature step spanning 2 h; then, the oven was turned off and the specimens were left to cool down in the oven (with the door shut) until the next day. For heating, an electrical 500 mm^3^ oven was used, with a maximum heating capacity of 1100 °C. This heating regime was adopted to allow for comparison with previous studies that followed the same protocol [32,33] and others varying only the ceiling temperature, which was kept constant for 1 h [34,35,36]. The specimens’ dimensions were measured with a vernier caliper, and the shrinkage/expansion was evaluated as the difference between pre- and post-high temperature exposure measurements. The specimens were also weighed before and after heating, using an electronic scale to calculate the mass loss. 

The position of the prisms in the oven was found to be quite important. Preliminary testing showed that placing the prisms in an upright position resulted in differential thermal cracking, with higher damage in the upper part of the specimens. Prisms placed closer to the resistances of the oven (in the preliminary phase) showed higher thermal damage. Based on these findings, it was decided to heat only four prisms at a time, which were symmetrically placed, as shown in Figure 1. All of the specimens were thus exposed to a similar thermal load. The prisms were placed on ceramic blocks to prevent overheating.

### 2.4. Testing of Mechanical Properties

The prisms were tested for flexural and compressive strength according to EN 1015-11. The loading rate was equal to 0.003 mm/s. For each mix, two prisms were tested for flexural strength before high thermal load exposure and one prism was tested after; all of the testing was performed under ambient conditions. The results in the form of average strength values are provided in a following section (average of two values of unheated flexural strength, four values of unheated compressive strength, and two test results of heated compressive strength). The mechanical properties assessed after heating and cooling down are typically lower than the corresponding values measured while the specimens are exposed to a high thermal load (higher than 800 °C) [37]. Thus, this methodology was deemed suitable for a conservative estimation of the compressive and flexural strength of heated low-calcium AAM products. The heated samples were removed from their plastic bags one day in advance to measure their dimensions and mass and were then placed in the oven for heat treatment. The mass and dimensions were measured again after cooling down to determine the shrinkage/expansion and mass loss.

### 2.5. XRD Method

Parts of the fired prisms (≈30 g) were milled for 30 s into a fine powder for X-ray powder diffraction with a Retsch RS200 vibratory ring mill (from Haan, Germany) at 1000 rpm. A bruker D2 phaser measured the diffraction patterns in the 5–70° 2q range with a 0.02° 2q and a step time of 0.6 s. CuKa radiation at 30 kV and 10 mA was used. The quantitative analysis was carried out using the Topas^®^ academic software V5 (created by Allan A. Coelho, Brisbane, Australia) [38]. 

### 2.6. Design of Experimental Matrix

The test matrix was designed with the aim of studying the effects of different proportioning scenarios of four dry components—namely FNS, KOH, KS (solid part), and SF—on selected wet and hardened properties of the pastes. For each binder blend, the sum of all the solid ingredients was kept equal to 3000 g, whereas 528 g of water was used to form paste. The water-to-binder (w/b) ratio was fixed to 0.176 based on preliminary trials (not described herein) to determine the lowest water-to-binder ratio that allowed for the formation of a workable alkali-activated paste. Table 3 shows the 34 combinations of the ingredients’ quantities (mixes) for each mix: (i) the components’ quantities have been scaled-up so that they sum to 1 ton of the dry binder, and (ii) the total water (accounting for the sum of mixing water and the liquid part of KS) is fixed at 176 kg so as to keep the w/b ratio at 0.176. The sum of all the ingredients is kept constant in order to isolate and study the effect of their proportions. This type of equality constraint is a prerequisite for the formation of a mixture DOE problem. 

The DOE matrix production and post-processing of the results were performed using Design Expert Software v11.1.2.0 (Stat-Ease Inc., Minneapolis, MN). Each component of the binder was assigned a letter to represent it: FNS = A, KOH = B, KS = C, and SF = D (see Table 2). As a first step to the design of the test matrix by DOE, single-component constraints were defined. The constraints for the lower and upper bounds of each ingredient (given in Table 2) aimed at achieving a certain range of potassium hydroxide and potassium silicate molar concentrations. The former was set to vary between 2 M and 7 M (the volume of reference being that of the chemical solution). Previous work on the activation of ferronickel slag [29] reported the attainment of high compressive strength when the KOH ranged between 4 M and 8 M and slightly lower strength at 2 M. The potassium silicate was set to vary from 0 M to 3.5 M. Higher molar concentrations were not possible due to the use of the commercial product Geosil 14517, which comes with a water content of 55%, thus making it impossible to increase the concentration of the silicate. The silica fume content varied between 0% and approximately 15% of the binder blend, in terms of weight.

An I-optimal mixture design that minimizes the average variance of prediction over the region of the test-derived results was used [39]. A special cubic Scheffe model was selected to account for three-component blending effects. The design comprised 34 runs: 14 design points were required to fit the model, whereas 10 additional points, 5 lack-of-fit points, 4 replicate points, and 1 additional center point were added as well in order to improve the prediction efficiency. Special care was given to keep the various work method processes (paste mixing and sampling) as consistent as possible between different runs: mixing time, vibrating time, and curing procedure. The experimentally evaluated responses (that is, wet and hardened paste properties) considered in the study were: shrinkage/expansion, mass loss, cost, and mechanical properties (flexural and compressive strength) before and after high thermal load exposure.

### 2.7. Life Cycle Assessment (LCA) Method

The LCA method was applied to estimate the environmental impacts of each FNS AAM formulation life cycle. Following the guidelines of ISO 14040, the LCA includes the following stages: goal and scope definition, life cycle inventory analysis, and environmental impact assessment and interpretation. To evaluate the environmental impacts associated with the developed mixes, a functional unit of 1 ton of the FNS AAM dry binder was defined. The system boundaries were limited to the production stage of the raw mix constituents in a “cradle-to-gate” framework”. As the AAM mixes were produced at ambient temperatures, the environmental impacts associated with the manufacturing of mixes are assumed to be similar across the specimens. The global warming potential (GWP) of the FNS AAM measured in CO_2_-equivalent emissions was estimated using the IPCC2013 method. The life cycle inventories (LCI) were collected from the literature, as well from the Ecoinvent v3.4 database accessible through the SimaPro 8.5 software. The consideration of the ferronickel slag as a waste material or industrial by-product determines the choice of the emission allocation method between the main process of ferronickel production and slag production through smelting in an electric arc furnace. For comparison, no allocation, mass allocation, and allocation based on economic value were examined. Additionally, the grinding of slag requires approximately 60 kWh/t at an industrial scale [40]. The environmental impacts associated with silica fume and potassium hydroxide production were estimated using Ecoinvent data. There is lack of data on the LCI of potassium silicate in the literature and in the LCI databases. Assuming similar production routes of the potassium and sodium silicates, we have estimated the environmental impacts of potassium silicate by using the LCI data of sodium silicate production through a hydrothermal process, where potassium hydroxide is used instead of sodium hydroxide as an input material.

## 3. Results and Discussion

The responses for all the DOE runs are provided in Table 4. These results were subsequently used for the derivation of regression equations (or models) in an effort to express each response (except for spread) as a function of the ingredients’ contents (see Section 3.1). Normal Plot, Residual Versus Predicted Plot, and Cook’s Distance were the tools used to identify the outliers for each response dataset. The outliers removed from a specific response dataset for the model derivation of this response are shown in Table 4 with a strike.

The maximum spread measured was that of the table diameter (300 mm). Mixes with a reported spread of 300 mm are likely to have a real spread higher than this value. Some replicate mixes, such as 28, 29, and 30, provided unexpectedly different results. The large variability could be reduced by producing more samples per mix for test purposes. Moreover, the variability in the results of the replicate mixes can be attributed to the inherent inconsistency of the precursor (no quality control over slag production, leading to varying chemical composition).

### 3.1. Derivation of Models 

Regression equations (or models) were derived for all responses (except for spread), and their coefficients are reported in Table 5. Equations for the prediction of each response can be obtained by multiplying the coefficients in Table 5 with the amount in grams of the respective ingredient (A: FNS, B: KOH, C: KS, D: SF, as provided in Table 3). It must be noted that the regression equations predict the properties of a hardened paste based on the combination of the ingredients A, B, C, and D and an amount of water suitable to keep the water-to-binder ratio at 0.176.

The presence of binary and tertiary coefficients (i.e., AB, ABC, etc.) indicates that nonlinear blending effects are relevant in this mix. The spread was measured as the diameter of the mortar sample after being jolted 15 times. The table diameter was 300 mm, which limited the maximum measured values. Several mix formulations spread beyond the flow table border, and the real spread diameter was not measured. Considering the lack of data for a big portion of the formulations, it was decided that the derived regression equation was not representative of this response.

Table 6 contains the mean response, the standard deviation (Std. Dev.), and the coefficient of variation (C.V.) of all the results for each response. The regression analysis was designed to fit a special Scheffe model. The final regression equations for each response were selected on an individual basis by utilizing an automatic model selection provided by Design Expert^®^ software. The criterion for term selection was to filter those that were significant for an associated significance level of 0.05 and simultaneously maximize the adjusted R^2^. The adjusted R^2^ was chosen to prevent the bias that comes with the pure R^2^, which is higher as more terms are added to the model, with the risk of overfitting the model by adding terms that are unrelated to the response.

For all the models, the difference between the predicted R^2^ and the adjusted R^2^ is less than 0.2, with the exception of the model that correlates ingredients’ contents and flexural strengths after high-temperature exposure. The high discrepancy between these two statistics indicates that the average value of heated flexural strength predicts the response better than the proposed model. All of the formulations subjected to high temperatures developed extensively distributed shrinkage/expansion cracking. Cracking damage had a remarkable impact on the flexural strength and promoted the early failure of all the samples. This early failure suggests that the real flexural strength potential of each formulation was not observed. It was hypothesized that preventing cracking of the prisms by utilizing fine aggregates or fibers would allow for higher heated flexural strengths to develop. A follow-up DOE in which fine aggregate and fibers were added to reduce thermal shrinkage/expansion followed this work and will be considered for publication. The adequate precision for all of the models was higher than 4, which is considered the minimum desirable ratio. This statistic measures the ratio of the signal (an effect in the response due to a change in the components) to the noise (random irregularities in the measured response). 

Contour and 3D surfaces were used to visualize the models and graphically study the impact of each component (total sum 3000 g). Figure 2a shows the heated compressive strength as a function of components A, B, and C, while component D was set to its minimum value (SF = 0). In this graph, the contour lines indicate an increase in strength as the content of component C (KS) increases. This suggests that removing silica fume from the mix results in a need for silicates to enhance the heated compressive strength.

In Figure 2b, the silica fume has been set to its maximum amount (SF = 441 g). Any more silicious oxide in the form of KS would result in the deterioration of the residual compressive strength. This is visible in the contour lines; as the lines get closer to the KS vertices (mixes with a higher content of KS), the strength decreases. The colors in the contour plot correspond to the amount of residual compressive strength. The blue color shows the zone where mix combinations including a high content of KS would result in low compressive strengths (around 5 MPa) compared to the high strength region displayed in green (15 to 20 MPa). The green area in the ternary plot contains the combination of three ingredients corresponding to a minimum value of KS and a higher concentration of KOH.

A similar visual analysis can be done in the 3D surfaces, which allows for the identification of local peaks more easily; such is the case for thermally induced volumetric change, as shown in Figure 3. The low level of silica fume resulted in shrinkage, with an average of -5%, as observed by the surface in Figure 3a. On the contrary, an excess of SiO2 for a scenario in which the silica fume is set to its upper bound yields varying volumetric change results (Figure 3b). The response surface model predicts that, at high silica fume contents (441 g), an increase in KS would result in a remarkable increase in volume. 

### 3.2. Optimal Paste Mix

Finally, after all the mix properties (with the exception of spread) were associated with a regression model (Table 5), it was possible to look for a combination of ingredients for an optimal formulation. The search of the “sweet spot” was executed numerically through the construction of a desirability function. To build the desirability function with the Design-Expert^®^ software, each response was assigned an importance factor and a criterion of optimization (minimization or maximization). Importance coefficients in Design-Expert^®^ are assigned on a scale from 1 to 5. The heated compressive strength was assigned the highest level, whereas the unheated compressive and flexural strengths were assigned level 4. All other responses were set at level 3. Heated flexural strength was assigned an importance of zero, as the regression model for this attribute was found to not be reliable due to the negative predicted R^2^. A negative R^2^ is produced when the average of the results is a better predictor of the response than the numerical model. By visual inspection of all the heated samples, it was clear that they showed extensive thermal cracking. As mentioned before, this induced premature flexural failure, making it unrealistic to try to correlate ingredients’ contents with residual flexural strength values.

The optimal dry binder composition is reported in Table 7 (ingredients adding up to 1 t). The performance markers reported in Table 7 correspond to a hardened paste prepared with a quantity of mixing water suitable to ensure a w/b ratio of 0.176. The optimal ingredients’ contents were found to be practically identical to those of Mix 13. Therefore, the measured responses of this mix were used for comparison with the predicted ones for the Optimal Mix proposed by Design-Expert^®^. The difference in performance markers (also given in Table 7) was found to be low for all responses (<9%), excluding the unheated compressive strength. The large error in prediction was anticipated, as the Predicted R2 of the unheated compressive strength model was the lowest—0.18 out of 1. It was expected before designing the experiment constraints that only a few maximum local points would be encountered. In the case of unheated compressive strength, the number of local optimal points is higher than those that can be captured by the models chosen while designing the experimental matrix. There are few combinations of ingredients that result in a compressive strength that is either too low (local minimums, suggested by the results measured from mix 4 (16.44 MPa), mix 5 (23.49 MPa), and mix 15 (13.88 MPa)) or too high (local maximums mix 13 (80 MPa), mix 25 (99 MPa) and mix 30 (94.5 MPa)). The lack of fit of these models could have been prevented if the lower and upper limits of the components would have been more restricted, thus resulting in a design space containing fewer local optimal points and in models with a higher level of prediction.

In the following sub-sections, the results presented in Table 4 are discussed per response. Additionally (and also per response), the optimal formulation is compared against both low-calcium alkali-activated pastes and cement-based ones. For the sake of a fair comparison, only works in which tests were executed at a thermal load between 800 °C and 1000 °C and kept constant for 1 to 2 h were considered. All values in the following bar charts correspond to samples cured at ambient temperature. Finally, the authors provide visual inspection notes on the specimens, along with the cost, LCA and XRD results, and relevant commentary.

### 3.3. Mass Loss

Mass loss was relatively consistent, varying from 12.5% to 17.5%. The amount of mixing water in each formulation was approximately 15%, by weight. A large part of the mass loss likely corresponded to the release of physical water (free water) between 20 °C and 100 °C, followed by the expulsion of chemically bonded water between 100 °C and 300 °C. Furthermore, between 200 °C and 650 °C, hydroxyl groups have been found to evaporate [41], adding to the total amount of mass loss due to water loss. The dehydroxylation of the Al-OH, Si-OH, and Ca-OH groups represents the second major reason for mass loss right after the evaporation of free water [42]. Nath et al. [43] and Rakhimova et al. [44] reported an additional mass loss after 750 °C due to the decomposition of carbonate groups.

A comparison of the mass loss for low-calcium alkali-activated and cement-based pastes after exposure to a thermal load is reported in Figure 4. The mass loss of the optimal formulation is highlighted in the figure with a darker color. Each bar label nomenclature corresponds to the main author and the year of publication of the bibliographic source, precursors (MK: Metakaolin, FA: Fly ash), and precursor fineness (d50, or specific surface). The figure shows that the FNS optimal mix performs equally well or better than cement-based pastes and much better than low-calcium alkali-activated ones.

### 3.4. Thermal Shrinkage/Expansion

The thermally induced volumetric change of the ferronickel mixes in this test campaign ranged from −8.3% (negative for shrinkage) to 21.7% (positive for expansion); the latter is atypical for alkali-activated materials, with the trend being a decrease in volume. After high thermal load exposure (900 °C), all of the prisms developed cracks of various intensities and patterns. In some cases, the cracks were abundant and wide, which resulted in an apparent increase in the size of the sample. The wider the cracks, the higher the apparent expansion value. Mineral phase transformation occurs as the temperature increases from the outside of the sample towards the core. The rate of transformation is thus not uniform throughout the cross-section of the specimen. The temporal incompatibility of the phases would result in differential stress throughout the cross-section, followed by crack formation. This is illustrated in Figure 5, which shows the expansion behavior of mix 31 to be an effect of the opening of cracks in the specimen. The appearance of cracks after high-temperature exposure is not unique for FNS AAM. Kong et al. [12] reported microcracks of 0.1–0.2 mm appearing in the surface of a metakaolin AAM after exposure to a thermal load of 800 °C. It has been observed in metakaolin [45] and fly ash [46] low-calcium AAMs that the addition of silica fume can increase the residual mechanical properties after a high thermal load by reducing the thermally induced volumetric change. A similar phenomenon was observed in FNS AAM. At low contents of KS, the increase in SF translates into a decrease in shrinkage. At high levels of KS, the shrinkage is not only reduced but reversed, and thermal expansion is recorded.

A comparison of the thermal volumetric behavior at the paste level for several low-calcium AAMs after exposure to a thermal load is reported in Figure 6. The thermal shrinkage of the optimal FNS mix (7%) is comparable to the shrinkage values reported by Zhang et al. (6.5%, [47]) and Rovnanik et al. (5%, [48]) for MK and FA AAM, respectively. The values are high compared to ordinary Portland cement paste, which shrinks down to around 1.7% at 800 °C [49]. It is possible to achieve low shrinkage geopolymer mixes, as Zhang et al. [47] demonstrated by studying the optimal combination of FA and MK, which yielded a thermal shrinkage of 1.05%, even lower than that of OPC. 

The intense shrinkage of low-calcium AAM pastes is a common characteristic of this family of materials. The thermal shrinkage of FNS corresponds to a typical behavior already documented by Rickard et al. [50] and Provis et al. [51], who found that alkali-activated materials with low calcium have an overall volumetric decrease produced mainly by the loss of free water between 100 and 300 °C, the dehydroxylation between 250–600 °C, and the densification by sintering typically between 550–900 °C. The volumetric contraction has been associated with an increase in the surface energy of the low-calcium AAM gel as water is released and a consequent partial collapse of the gel network takes place [41]. Bernal et al. [52] proposed that this partial collapse would likely induce damage in the form of microcracks, which would result in an overall decrease in strength. These microcracks may be responsible for the strength drop of the optimal mix from 80 MPa to 16 MPa. Bakharev [53] proposed that volumetric stability should be a fundamental characteristic of a fire-resistant low-calcium AAM. The reduction of the considerable shrinkage of FNS AAM represents a challenge that will be undertaken in future studies.

### 3.5. Compressive Strength

The compressive strength before high thermal load (900 °C) exposure ranged between 12.28 MPa and 99.6 MPa, with an average of 51.2 MPa. Runs 13, 14, 25, and 30 are of particular interest due to their high compressive strength exceeding 80 MPa. From these mixes, runs 26 and 31 are unsuccessful due to excessive thermal cracking, which is similar to what is pictured in Figure 5. Runs 13 and 14 have approximately the same composition, with mix 14 having a slightly higher content of silicates. The latter showed a higher compressive strength value, suggesting a correlation between silicate content and strength development. A similar trend was observed in runs 26 and 31 which had a considerably high amount of silicate in the form of both SF and KS. Runs 4, 5, 15, 17, 18, and 33 have a compressive strength under 25 MPa. These mixes were designed with zero potassium silicate contents, which again highlights the importance of silicates as a key ingredient for strength development. 

Si/Al ratios have previously been associated with the strength of low-calcium AAM both before and after high-temperature exposure [12,54]. Increasing the Si/Al ratio implies more Si-O-Si bonds that are stronger when compared to the Si-O-Al ones and would thus result in a higher (unheated) strength [55]. In this study, such correlation was not found, as depicted in Figure 7.

The calculated coefficients of determination between the Si/Al ratios and compressive strengths for ferronickel slag low-calcium AAM were found to lie below 0.22 and indicate very poor correspondence. The lack of correlation between the compressive strength and Si/Al ratio has also been reported in other studies [35]. The addition of silica fume in the mixture likely had a strong influence on the mechanical properties of the mix, which obscures the role that the Si/Al ratio has on the measured responses. Additionally, the range of the Si/Al ratio studied herein (2.8 to 4.3) does not coincide with that in studies reporting a correlation between Si/Al and compressive strength (see Kong et al. [12] (1.4 to 2.3) and Duxson, Lukey, & van Deventer [54] (1.15 to 2.15)). Finally, the Si/Al ratios in this study were calculated based on the assumption that the oxides of silicon and aluminum have fully reacted in the AAM matrix. This is unlikely, since there will be a portion of Si and Al containing particles that are undissolved after the matrix formation [56].

Several studies indicate that an increase in compressive strength is linked to a higher concentration of the alkaline activator (NaOH and KOH) [57,58,59]. This trend has been confirmed in further studies, indicating that the trend is nonlinear. Nevertheless, it has been observed that, at high levels of the alkaline activator, a strength drop occurs. The previously monotonous relationship becomes non-linear [60,61]. Previous studies on ferronickel slag-based low-calcium AAM have confirmed the latter observation, and the inflection point was found at a NaOH molar concentration of 7M. In this study, the nonlinear behavior becomes even more complex due to the addition of silica fume. The linear and even the nonlinear correlation become less evident, as some mixes with a low chemical concentration showed some of the highest values of compressive strength. Mix 13, 14, and 25 with a KOH concentration of 2M had compressive strength values after 42 days of 80 MPa, 85 MPa, and 99 MPa respectively, while there were cases of mixes with moderately high KOH concentrations (7M), which resulted in low strengths (mixes 12 and 15, which reported 16.7 MPa and 13.9 MPa, respectively). The nonlinear interaction of ingredients brings forth the possibility of achieving high mechanical properties with low chemical concentrations, which improves the sustainability, safety of use, and economic feasibility of ferronickel slag low-calcium AAM.

After heating, the visible reduction in strength is a consequence of phase transformations and damage produced by pore pressure effects [62,63]. Phase transformations have been reported to result in large cracks and a consequent strength reduction in fly ash-based low-calcium AAM [64,65]. Large cracks were also observed in FNS-based low-calcium AAM, which suggests a similar damage mechanism. The compressive strength after a high thermal load varied from 2.8 to 37.5 MPa. Mixes 20, 21, and 23 showed the highest values of residual compressive strength after HTL exposure. These runs had almost the same content of SF. It must be noted that some of the mixes have a high residual compressive strength despite the high iron content, which has been previously correlated with negative effects on the high-temperature performance of low-calcium AAM [50,66]. 

Previous works (between 800 °C and 1000 °C) associated lower Si/Al ratios with higher residual strengths [14,35]. On the contrary, other authors like Kong [12] correlate higher Si/Al ratios with smaller reductions in strength after a high thermal load, or even with increases in residual strength [67]. Despite this, while not observable in the present FNS AAM (see Figure 7), a vast amount of literature indicates that the Si/Al ratio is a critical parameter that affects the mechanical properties and phase formations of low-calcium AAM materials after high-temperature exposure [53,68,69].

Authors such as Rashad & Zeedan [70] studied the effect of increasing chemical activator content in low-calcium fly ash AAM exposed to a range of temperatures between 200 °C and 1000 °C. The authors found that an increased content of waterglass resulted in a further decrease in residual compressive strength. Once again, this trend was not found in the ferronickel slag low-calcium AAM developed in this investigation. Likely, the reason for this is the upper limit of the KOH concentration (7 M), which is low when compared with concentrations studied by other authors, reaching 10 M. In the range studied herein (2 M to 7 M), an increase in KOH was found to improve residual compressive strength. Previous studies on FNS found higher mechanical strength values with lower NAOH concentrations—specifically 6 M and 8 M, with a better performance than low-calcium AAM prepared with 10 M and 12 M [26]. Komnitsas et al. [29] reported that an excess of 10M KOH resulted in decreased strength for FNS AAMs.

Mixes 5 and 4 resulted in a residual compressive value of 10.31 MPa and 19.54 MPa. The only difference in composition is an increase in the KOH concentration from 2 M in Mix 5 to 7 M in Mix 4. While these two formulations included no silica fume, the same trend was observed in Mixes 32 and 33, where the residual compressive strength grew from 6.8 to 25.4 MPa as the KOH concentration increased from 2 M to 7 M.

The compressive strength of the binder after exposure to a high thermal load (900 °C) ranked above average in the comparison chart presented in Figure 8. Again, the figure includes only results from studies not employing heat curing. All the pastes in Figure 8 had unheated strengths of at least 35 MPa (hence, they could be used as the paste phase in a prospect concrete formulation). The high compressive strength of the FNS paste (darker color in Figure 8) is partially provided by an optimal percentage of silica fume. Sivasakthi [71] studied the effect of silica fume on low-calcium AAM mixes from 0 to 10% and found the highest gain at 5%, not far away from the 6% value obtained in this study. 

By expressing residual compressive strength as a percentage of the initial (unheated) one, it can be seen that both alkali-activated binders in Figure 8 are similar (14% [52] and 20% for FNS), whereas the cement-based ones result in zero or very low residual strengths [3,11,72]. The strength before high-temperature exposure was found to be one of the highest among the low-calcium AAM binder results compiled. High strength, as reported by Duxson et al. [73], is a consequence of the chemical bonds formed in the aluminosilicate gel and the physicochemical interaction between the unreacted particles and the gel. 

As mentioned before, FNS AAMs showed extensive damage due to thermal cracking; thus, the authors believe that the mechanical strength of the FNS binder in compression after a high thermal load could be much higher if the cracking is controlled with fibers or by using fine fillers. 

### 3.6. Flexural Strength

Runs 13 and 14 showed the highest unheated flexural strength of all the combinations (6.7 MPa and 7.3 MPa, respectively). In general, most mixes (65%) have an unheated flexural strength of 3 MPa or higher. The residual flexural strength after exposure to a high thermal load (900 °C) was found to be low in all specimens. The strength loss is probably a result of the cracking damage suffered during the heating phase. The cracks substantially affected the flexural strength by reducing the effective cross-section of the specimen. Thermal expansion has been associated with cracking in metakaolin low-calcium AAM and is also likely the reason for the occurrence of cracks in FNS AAM (also supported in [56]). 

The flexural strength before exposure to a high temperature was found to be comparable to fly ash and metakaolin alkali-activated materials, as shown in Figure 9. Values were selected for comparison based on a criterion of a minimum flexural strength of 5 MPa. Only non-heat-cured geopolymer results were included. The strength of FNS after HTL exposure was zero due to the cracks formed in the prismatic specimens. Similar results were reported for the OPC samples [3,72]. The MK mix by Rovnanik [48] barely yielded any strength after high temperature exposure (0.2 MPa), while the work of Kovarik showed an outstanding residual flexural strength using MK as precursor [74].

As was mentioned in the previous section, there is a poor correlation between the Si/Al ratios and the compressive strengths of the FNS AAMs produced in this study. While this is likely due to the interference of silica fume and its double role as both a source of silica and a micro filler, there is another parameter that, while less popular in the literature, is relevant for the analysis of low-calcium AAM at a high temperature and K/Al ratio. Kohout et al. [36] studied the relationship between the K/Al ratios and post-fired properties of low-calcium alkali-activated systems. The author recommended keeping materials intended for fire resistance between a K/Al ratio of 0.55 and 0.70. Remarkably, the optimal mix produced by the DOE design of mixtures has a K/Al ratio of 0.66, falling right in the middle of this desirable range.

### 3.7. Appearance

In Figure 5, the change in color from greenish to red can be noticed. This phenomenon has also previously been observed in other low-calcium alkali-activated materials and is usually associated with the oxidation of the iron species [10,50,75], which have been confirmed to exist in abundance in ferronickel slag.

### 3.8. Cost

The authors found no correlation between (either unheated or residual) flexural or compressive strength and cost. This lack of causation indicates that the formulation can be optimized to increase the mechanical properties without directly implying an increase in cost. The same behavior was observed for the residual mechanical properties after a high thermal load. Figure 10 shows the scattered plots of mechanical properties as a function of cost. Notice the values of R^2^, which, in all cases, fall below 0.124.

### 3.9. Life Cycle Analysis

The GWP of the examined mixes is presented in Figure 11. The activator contributes the most towards the GWP of the analyzed mixes. Potassium silicate is responsible for up to 69% of the total CO_2 eq_. emissions (mix 31). The GWP of mix 13 is 99 kg, where the activator contributes 76% and FNS 31%.

When FNS is classified as a waste material—and therefore only emissions attributable to the grinding are accounted for—the total environmental impacts of the AAM are the lowest compared to the economic allocation. In this study, the economic allocation is based on the ferronickel price of EUR 11,036/t [76] and of the FNS of EUR 31/t. Mix 13 is compared to 1 ton of OPC, as a similar mechanical performance can be expected by using these materials [19]. The production of 1 ton of OPC is associated with 870 kg CO_2_ eq. However, according to the environmental product declarations, there is a variability in the reported environmental impacts across cement production plants [77], which is illustrated in Figure 12. The GWP of Mix 13 is 99 kg CO_2 eq_., whereas, considering the economic allocation approach, it is 388 kg CO_2_ eq. In comparison with the average OPC impact, the environmental footprint of Mix 13 is 89% lower when FNS is considered as a waste material and remains 55% lower when FNS is considered as a by-product and economic allocation is applied (Figure 12).

### 3.10. XRD

The main crystalline phases for all samples (Figure 13) were magnetite (Fe_3_O_4_) and the pyroxene solid-solution (Ca,Mg,Fe)_2_(Al,Si,Fe)_2_O_6_, which crystallizes from the slag and binder, as iron-containing glasses are prone to crystallization, especially in air. Leucite (KAlSi_2_O_6_) crystallized for the samples with a lower SiO_2_/K_2_O molar ratio (i.e., samples with a higher addition of the alkaline solution Fayalite ((Mg,Fe)_2_SiO_4_) was also found to be a minor crystalline compound in the samples (except in sample 13, where it was the major phase). Fayalite crystallizes from the unreacted parent slag and binder phase and decomposes further to hematite/magnetite and silica in the air [78]. Additionally, hematite (Fe_2_O_3_) was present in most of the samples, which is the stable form of the iron oxide phase at this temperature; however, the porosity and microstructure of the sample will determine the oxidation rate of fayalite and magnetite, which crystallizes initially from the binder. 

No correlations between the mechanical performance after firing and the phase assemblage were found. The mechanical strength after firing depends more on the cracks/defects forming during firing as well as on how well the samples densify (porosity after firing) than it does on the specific phase assemblage formed after firing.

## 4. Conclusions

The Design of Experiment was proven to be useful in finding the combination of components (SF, FNS, KS, KOH) that enhanced the mechanical properties before and after firing and simultaneously minimized the chemical activator content. This resulted in a low CO_2_ emissions recipe for AAM for high temperature applications.

The mass loss of ferronickel slag low-calcium AAM activated by a potassium silicate and potassium hydroxide solution has shown a similar or lower value compared to well-known metakaolin and OPC formulations while still underperforming when compared to fly ash alkali-activated materials.

The thermal shrinkage/expansion of the FNS optimal formulation was high in comparison to OPC and the other AAM. The shrinkage/expansion might be reduced by the addition of fine aggregates, which requires a study that will follow this paper.

KOH and KS are fundamental parameters that are necessary to fine-tune a low-calcium AAM mix for high temperature applications. The optimal mix design was optimized to minimize chemical activators, and it was found that FNS can be activated with molar concentrations as low as 2 M KOH and 1.36 M KS to produce compressive and flexural strengths as high as 80 MPa and 6.8 MPa, respectively. 

Silica fume has an important role in improving the thermal performance of FNS-based low-calcium AAM; the optimal ratio was found to be 6%.

The DOE proved to be a powerful tool for low-calcium AAM mix design, as the optimal formulation not only resulted in a high strength but also in the minimization of cost and environmental impact.

There is no strong correlation between mechanical properties such as flexural and compressive strength and the cost of the formulation. This implies that improving the performance of a low-calcium AAM formulation does not necessarily translate to a higher cost.

The optimal formulation would cut CO_2_ emissions by up to 55% if FNS is considered as a by-product and by up to 89% if FNS is categorized as a waste. In addition to this reduction in the manufacturing, there is a reduction in the environmental impact, as the slag would not be allocated in landfills. 

It is possible to produce a geopolymer mix for ambient and high temperature applications with a low concentration of chemical activators and without the need for heat curing. This reduces the environmental and safety cost of utilizing AAM in the construction industry. 

## Figures and Tables

**Figure 1 materials-15-04379-f001:**
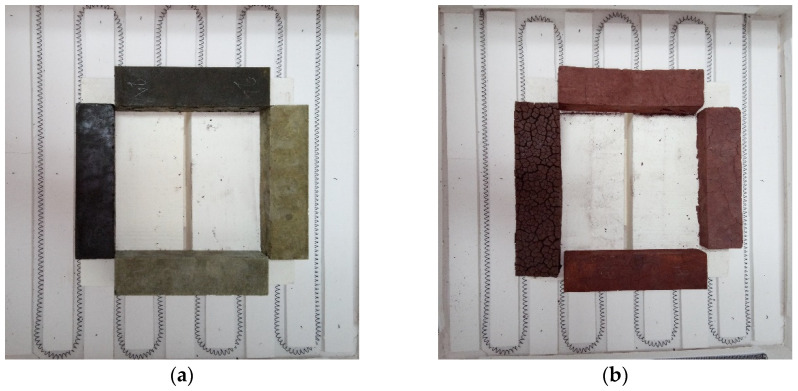
Specimens of different compositions placed in the oven: (**a**) before and (**b**) after thermal load application.

**Figure 2 materials-15-04379-f002:**
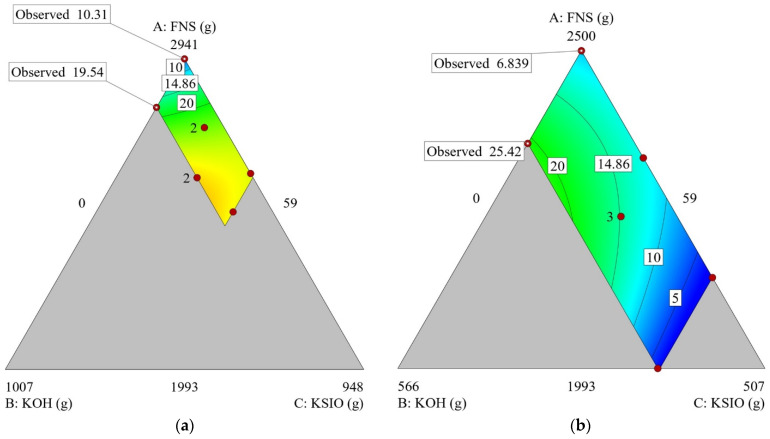
Example contour graphs of post-heating residual compressive strengths as a function of the contents of components A, B, and C, with component D fixed to its: (**a**) lower bound (0 g), (**b**) upper bound (441 g). The mix sums to 3000 g.

**Figure 3 materials-15-04379-f003:**
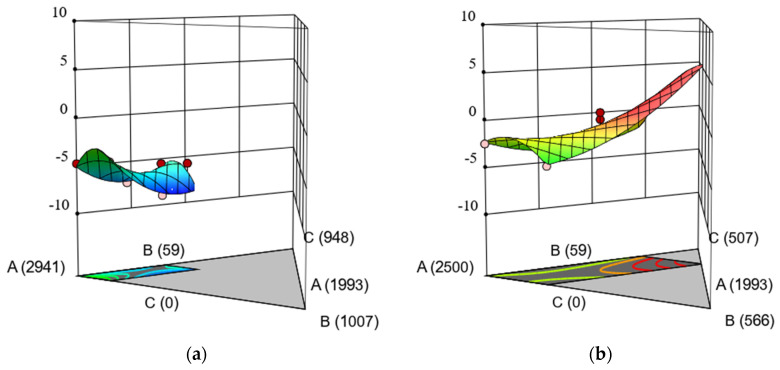
3D response surface model (RSM) of thermally induced volumetric change as a function of the contents of components A, B, and C, with component D fixed to its: (**a**) lower bound (0 g), (**b**) upper bound (441 g). The mix sums to 3000 g.

**Figure 4 materials-15-04379-f004:**
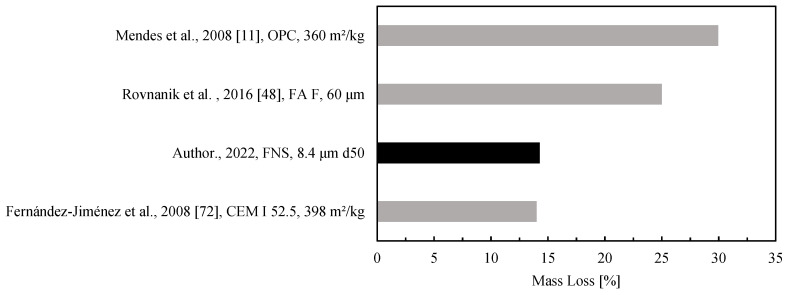
Mass loss of low-calcium alkali-activated and cement-based pastes after exposure to a thermal load between 800 and 1000 °C.

**Figure 5 materials-15-04379-f005:**
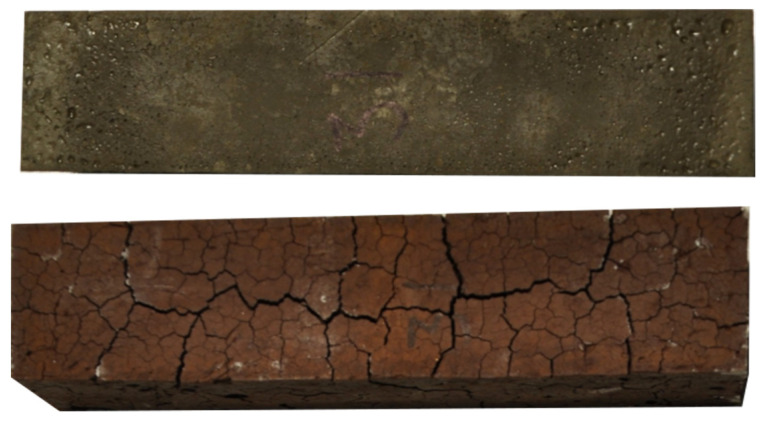
Top view of sample 31: up, before heat exposure; down, after heat exposure.

**Figure 6 materials-15-04379-f006:**
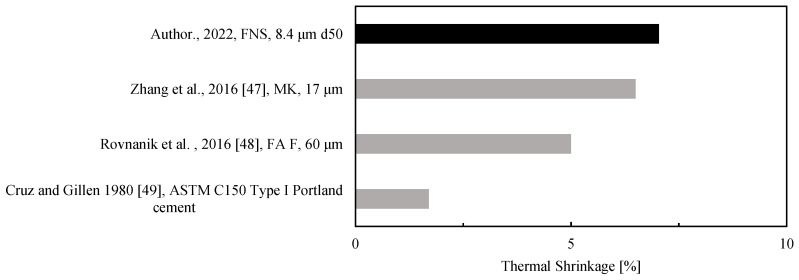
Thermal shrinkage of low-calcium AAM pastes after exposure to a thermal load between 800 and 1000 °C.

**Figure 7 materials-15-04379-f007:**
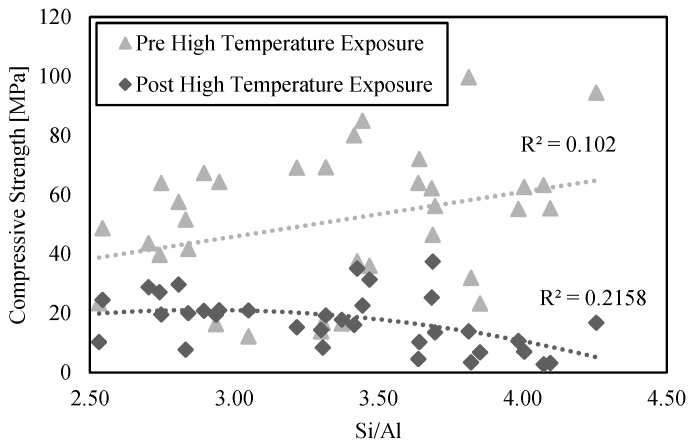
Compressive strength before and after the high thermal load as a function of the Si/Al ratio.

**Figure 8 materials-15-04379-f008:**
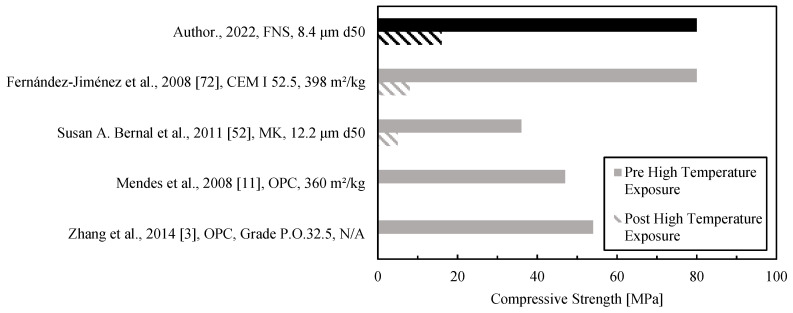
Compressive strength of low-calcium AAM pastes after exposure to a thermal load between 800 and 1000 °C.

**Figure 9 materials-15-04379-f009:**
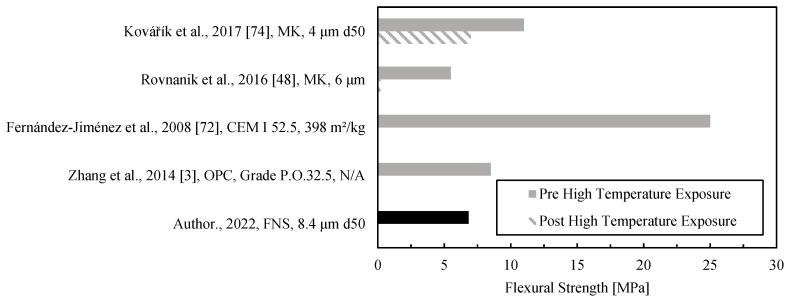
Flexural strength of low-calcium AAM pastes after exposure to a thermal load between 800 and 1000 °C.

**Figure 10 materials-15-04379-f010:**
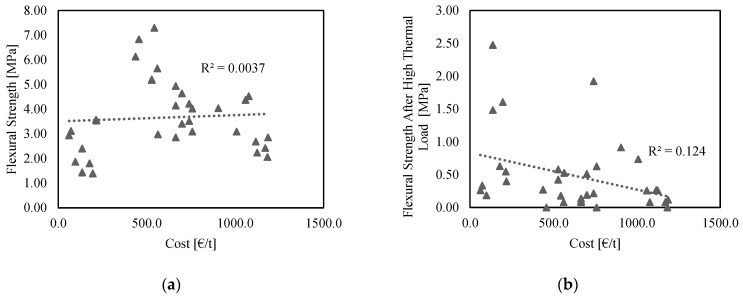
Correlation between cost and: (**a**) unheated flexural strength; (**b**) flexural strength after a high thermal load; (**c**) unheated compressive strength; (**d**) compressive strength after a high thermal load. For all samples, the water-to-binder ratio equals 0.176, and the tests occurred after 42 days of curing in ambient conditions, preventing moisture loss.

**Figure 11 materials-15-04379-f011:**
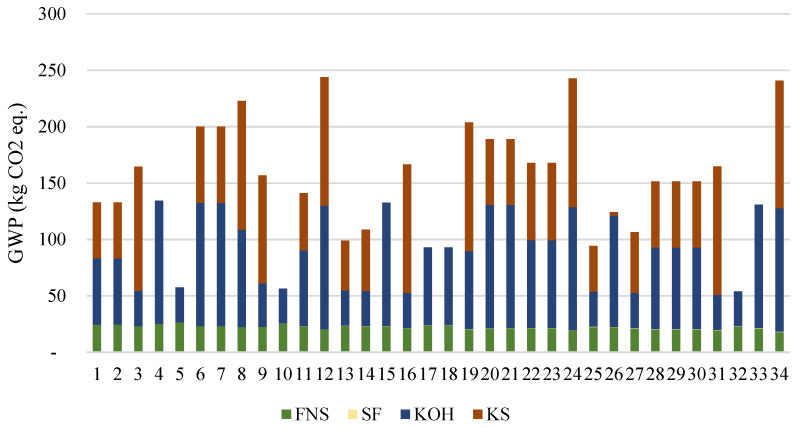
GWP of the analyzed mixes.

**Figure 12 materials-15-04379-f012:**
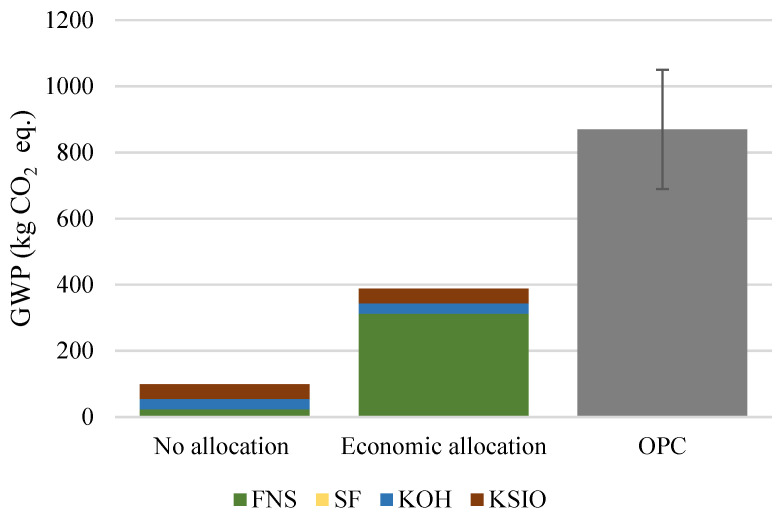
Impact of the allocation method on the total GWP of FNS AAM compared to OPC.

**Figure 13 materials-15-04379-f013:**
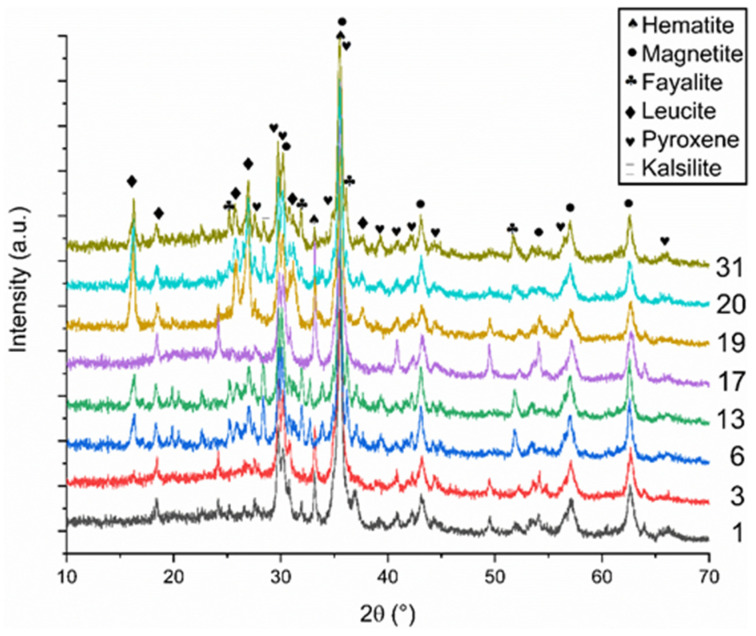
X-ray diffractogram of the samples of run 1, 3, 6, 13, 17, 19, 20, and 30 after firing to 900 °C for 2 h.

**Table 1 materials-15-04379-t001:** Ferronickel slag and silica fume chemical analysis through XRF (Mass/%) *.

Composition	SiO_2_	Al_2_O_3_	CaO	Fe_2_O_3_	MgO	Na_2_O	P_2_O_5_	K_2_O	TiO_2_	MnO	LOI
Ferronickel	36.9	3.61	4.18	32.8	7.41	0.15	0.02	0.48	0.19	0.00	0.00
Silica fume	88.9	0.73	0.34	1.01	0.63	0.71	0.03	1.50	0.00	0.12	6.82

* Only detectable chemical compounds are listed.

**Table 2 materials-15-04379-t002:** Cost, CO_2-eq._, energy consumption, and DOE boundaries for each binder ingredient.

Material	Code	Cost (Euro/t)	CO_2-eq_. (kg/t)	Energy Consumption (MWh/t)	Mixture Boundaries by Weight (%)	
					Lower Bound	Upper Bound
FNS	A	31 *	26.9 **	0.06	66.4	98.0
KOH	B	5000	945.9	3.9	2.0	6.9
KS	C	17,089	1585.3	0.3	0.0	12.1
SF	D	5500	3.1	0 ***	0.0	14.7

* Cost of ground slag; ** Based on energy consumption during grinding; *** Silica fume is considered a waste material in Ecoinvent v3.4.

**Table 3 materials-15-04379-t003:** Ferronickel slag-based alkali-activated paste compositions for 1 m^3^ (176 kg of water was added to each mix).

Mix Number	Build Type	Space Type	FNS (kg)	SF (kg)	KOH (kg)	KS (kg)
1	Model	Plane	910.47	0.00	37.05	52.48
2	Replicate	Plane	910.47	0.00	37.05	52.48
3	Model	Edge	863.76	0.00	19.67	116.57
4	Model	Vertex	931.00	0.00	69.00	0.00
5	Model	Vertex	980.33	0.00	19.67	0.00
6	Model	Edge	859.48	0.00	69.00	71.52
7	Replicate	Edge	859.48	0.00	69.00	71.52
8	Model	Edge	824.68	0.00	54.65	120.67
9	Lack of Fit	Interior	832.50	41.97	24.27	101.26
10	Lack of Fit	Edge	934.59	45.74	19.67	0.00
11	Model	Interior	853.25	50.57	42.43	53.75
12	Model	Edge	758.46	51.87	69.00	120.67
13	Model	Plane	870.22	63.30	19.67	46.81
14	Model	Plane	857.41	65.56	19.67	57.37
15	Model	Edge	859.96	71.04	69.00	0.00
16	Model	Edge	786.66	73.01	19.67	120.67
17	Model	Plane	882.18	74.32	43.50	0.00
18	Model	Plane	882.18	74.32	43.50	0.00
19	Replicate	Plane	760.28	75.48	43.57	120.67
20	Model	Plane	782.08	87.29	69.00	61.64
21	Model	Plane	782.08	87.29	69.00	61.64
22	Center	Center	790.43	88.10	49.17	72.30
23	Replicate	Center	790.43	88.10	49.17	72.30
24	Lack of Fit	Edge	709.50	100.83	69.00	120.67
25	Lack of Fit	Plane	826.96	110.46	19.67	42.91
26	Lack of Fit	Interior	823.59	110.80	62.17	3.44
27	Model	Edge	776.29	147.00	19.67	57.04
28	Model	Plane	745.16	147.00	45.53	62.32
29	Model	Plane	745.16	147.00	45.53	62.32
30	Replicate	Plane	745.16	147.00	45.53	62.32
31	Model	Vertex	712.67	147.00	19.67	120.67
32	Model	Vertex	833.33	147.00	19.67	0.00
33	Model	Vertex	784.00	147.00	69.00	0.00
34	Model	Vertex	664.33	147.00	69.00	119.67

**Table 4 materials-15-04379-t004:** Experimentally evaluated responses of ferronickel slag-based alkali-activated pastes.

Mix Number	Spread (mm)	Mass Loss (%)	Shrinkage/Expansion (%)	Flexural Strength (MPa)		Compressive Strength (MPa)		Cost (€/t)
				HTL *	Unheated	HTL	Unheated	
1	300	14.6%	−6.0%	0.58	5.18	19.61	64.02	528.7
2	300	14.6%	−7.0%	0.43	5.21	21.09	64.31	528.7
3	300	15.0%	−5.5%	0.74	3.09	24.60	48.76	1008.2
4	170	13.0%	−4.2%	0.63	1.80	19.54	16.44	178.0
5	100	12.7%	−4.8%	0.26	2.93	10.31	23.49	61.3
6	300	17.5%	−8.3%	0.63	4.03	28.90	43.64	758.9
7	300	15.8%	−7.50%	0.00	3.09	27.12	39.76	758.9
8	300	13.7%	−5.26%	0.26	2.24	29.78	57.56	1124.2
9	300	16.2%	−7.48%	0.92	4.04	20.13	41.71	905.1
10	160	12.5%	−6.75%	0.34	3.11	7.80	51.58	72.6
11	300	16.6%	−6.03%	0.53	2.98	20.91	67.37	564.3
12	300	17.1%	−5.71%	0.08	2.42	17.87	16.66	1171.0
13	280	14.3%	−7.04%	0.00	6.83	16.00	80.00	457.2
14	290	15.4%	−6.27%	0.18	7.30	22.71	85.00	543.5
15	200	13.9%	−5.00%	1.61	1.39	14.42	13.88	195.6
16	300	16.4%	−6.00%	0.26	4.38	19.36	69.27	1059.5
17	170	13.2%	−4.73%	1.48	2.40	8.41	16.80	136.1
18	190	12.8%	−3.25%	2.48	1.44	21.04	12.28	136.1
19	300	13.9%	5.99%	0.27	2.68	10.30	72.06	1116.7
20	300	16.2%	−6.25%	0.19	4.64	37.48	46.43	700.3
21	300	16.3%	−5.24%	0.52	3.40	31.47	36.02	700.3
22	300	15.0%	0.75%	0.22	4.22	15.24	69.11	740.2
23	300	16.3%	−6.27%	1.92	3.51	35.10	37.60	740.2
24	300	13.7%	21.70%	0.00	2.06	4.62	64.00	1183.1
25	250	16.9%	−6.27%	0.27	6.13	13.88	99.61	437.1
26	190	13.7%	−6.50%	0.40	3.54	3.50	32.00	217.2
27	260	16.4%	−4.50%	0.08	5.65	13.62	56.25	560.9
28	300	16.1%	4.26%	0.08	4.15	7.05	62.67	665.0
29	300	17.0%	−0.25%	0.10	4.94	10.71	55.20	665.0
30	300	14.2%	0.50%	0.14	2.85	16.82	94.51	665.0
31	300	17.2%	12.50%	0.08	4.53	2.79	63.34	1077.8
32	140	13.9%	−2.49%	0.19	1.86	6.84	23.33	97.6
33	210	16.1%	−4.25%	0.55	3.57	25.42	62.16	214.4
34	300	14.8%	11.75%	0.12	2.85	3.21	55.46	1186.4

* High thermal load.

**Table 5 materials-15-04379-t005:** Actual model-coded coefficients of responses.

Response	A	B	C	D	AB	AC	AD	BC	BD	CD	ABC	BCD
Mass loss (%)	0.0043846	0.4764324	−0.116051	0.0108843	−0.000168	5.033 × 10^−5^		−0.000158	−0.000187	4.122 × 10^−5^		
Shrinkage/expansion (%)	−0.002897	−0.715021	0.1377021	0.1209094	0.0002632	−5.24 × 10^−5^	−4.49 × 10^−5^	0.0001457	0.0001736	−0.000122		5.288 × 10^−7^
Flexural strength unheated (MPa)	0.0018497	0.256696	−0.327099	−0.005653	−9.83 × 10^−5^	0.0001257		0.0010853	−4.1 × 10^−5^	0.0001593	−4.32 × 10^−7^	−6.74 × 10^−7^
Flexural strength heated (MPa)	0.0002595	0.0012419	−0.001349	−0.000349								
Compressive strength unheated (MPa)	0.0108865	−0.085398	−0.394636	0.0828144		0.0002072						
Compressive strength heated (MPa)	0.0006383	0.0750318	−0.416928	0.0082899								

**Table 6 materials-15-04379-t006:** Fit statistics for models.

Response	Mean	Std. Dev.	C.V. %	R^2^	Adjusted R^2^	Predicted R^2^	Adequate Precision
Mass loss (%)	15.09	0.98	6.51	0.67	0.56	0.31	9.93
Shrinkage/expansion (%)	−5.30	1.04	19.61	0.83	0.73	0.61	11.06
Flexural strength unheated (MPa)	3.66	0.74	20.22	0.82	0.75	0.63	11.99
Flexural strength heated (MPa)	0.49	0.55	113.25	0.17	0.09	−0.01	5.28
Compressive strength unheated (MPa)	51.24	19.28	37.63	0.39	0.31	0.18	8.14
Compressive strength heated (MPa)	17.28	6.46	37.35	0.58	0.52	0.46	11.46

**Table 7 materials-15-04379-t007:** Optimized formulations and error in predictions.

Response	FNS (kg)	SF (kg)	KOH (kg)	KS (kg)	Mass Loss (%)	Shrinkage (%)	Flexural Strength Unheated (MPa)	Compressive Strength Heated (MPa)	Compressive Strength Unheated (MPa)	Cost (€/t)
Predicted	870.33	63.30	19.67	46.67	15.62	−7.66	6.28	16.93	59.65	456
Measured (Mix 13)	870.22	63.30	19.67	46.81	14.28	−7.04	6.83	16.00	80.00	457
Difference (%)	−0.01	0.00	0.00	0.31	−8.58	−8.09	8.81	−5.49	34.12	0.22

## Data Availability

The data presented in this study are available on request from the corresponding author.

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
