# Peer review of "Optimal Design of Ferronickel Slag Alkali-Activated Material for High Thermal Load Applications Developed by Design of Experiment"

_materials, 2022, doi:10.3390/ma15134379_

Round 1

Reviewer 1 Report

The paper “Optimal design of ferronickel slag alkali-activated material for high thermal load applications developed by design of experiment” deals with the optimization of low-calcium alkali-activated binder for high-temperature stability as a more sustainable alternative of Ordinary Portland Cements (OPC).

To this Aim, Arce et al. aim to find the proportions of ingredients (i.e., ferronickel slag, potassium hydroxide, potassium silicate, silica fume and water) in order to optimize properties such as flow spread, mass loss, shrinkage/expansion, compressive and flexural strength before and after a high thermal load (i.e., 900 °C for 2 hours) as well as the CO2 emissions and a cost of the binder formula.

In my opinion, this paper is well written and organized, the results sound good and its topic is certainly interesting for the Readers of Materials. Nevertheless, some revision is needed to improve the quality of the paper.

In the following my detailed comments:

- Line 143: Please remove the first sentence of section 2.1

- Lines 150 and 163: please remove the sentences “at the Laboratory of Sedimentology of the University of Patras” and “the Laboratory of Electron Microscopy and Microanalysis of the University of Patras”;

- Line 233: Are the Authors sure that only one mechanical test is enough to evaluate the flexural properties of the binder after the thermal load exposure? Moreover, they stated that “results in the form of average strength values are provided”. How can they talk about average strength if two or one flexural tests were performed on prismatic samples (i.e., before and after the thermal load exposure)?

- Line 342: From my point of view, the sentence “the actual spread values of these mixes were higher than 300 mm and were not captured by the test” since it is a repetition;

-  Line 382: The Authors are encouraged to better explain the statement “The blue area (5 MPa strength region) in this figure also suggests that an excess of silica would result in a meaningful drop (compared to 15-20 MPa in the green region) in this response”;

- Line 420: Why all the differences between the measured responses with the predicted ones for the optimal mix proposed by Design-Expert tool were found to be lower than 9% apart from the unheated compressive strength? The Authors should add a convincing explanation to this finding instead of listing it;

- Line 458: By checking Table 4, it seems that the minimum volumetric change experienced by the ferronickel mixes was -7.5% (Mix 7) instead of -8.3%, as reported in the text;

- Line 463: The Authors should better explain their statement “Non-uniform distribution of alkali activated compounds 463 throughout each specimen might also add to the cracking potential of heated prisms”;

-  Line 504: By checking Table 4, it seems that compressive strength before high thermal load exposure ranged between 12.28 MPa (mix 18) and 99.61 MPa (mix 25) instead of 23.3 MPa and 99.6 MPa, as reported in the text. Furthermore, the compressive strength of mix 26 and 31 are lower than 80 MPa. Please correct these errors in the text;

- Line 530: The meaning of the sentence “Si/Al ratios in this study correspond to a complete reaction of the precursors and activators which is not guaranteed”;

- Line 595: “heated” should be replaced with “residual”;

- Line 698: I suggest to add an introducing sentence at the beginning of the Conclusions section to emphasize the main objective of the present paper.

Author Response

Thank you for taking the time and detail to go so punctually over the manuscript. Your corrections are very valuable to us. 

For our reply, please see the attachment.

Reviewer 2 Report

Too little testing of mechanical properties, especially depending on the exposure temperature and depending on the material composition and concentration. No structure research.

Author Response

Thank you for taking time out of your busy day to review the manuscript.

For our reply, please see the attachment.

Reviewer 3 Report

This paper introduced an optimal design of ferronickel slag alkali-activated material. In view of solid waste utilization, it is significant. But in view of innovativeness, it seems lack of necessary innovativeness and scientific contents. It looks like a research report on an alkali-activated material using a slag as main raw material and focuses on talking about those test results, not explain the reasons. So I suggest it is rejected.

Author Response

Thank you for taking time out of your busy day to review the manuscript. We understand the manuscript at a first impression does not seem suitable for publication.

I understand that your expertise has brought you to this conclusion. I am sure your experience in the field is long, and by no means do I intend to contradict your conclusion about the paper. This reply intends only to provide a different angle to a manuscript that might be valuable in your eyes when observed from a different perspective. 

Please see the attachment for our full reply.

Round 2

Reviewer 2 Report

I I accept the article in presented form.